# Altered Pain Processing Associated with Administration of Dopamine Agonist and Antagonist in Healthy Volunteers

**DOI:** 10.3390/brainsci12030351

**Published:** 2022-03-04

**Authors:** Sarah L. Martin, Anthony K. P. Jones, Christopher A. Brown, Christopher Kobylecki, Grace A. Whitaker, Wael El-Deredy, Monty A. Silverdale

**Affiliations:** 1Department of Psychology, Manchester Metropolitan University, Manchester M15 6GX, UK; 2The Human Pain Research Group, Division of Neuroscience and Experimental Psychology, The University of Manchester, Manchester M13 9PL, UK; anthony.jones@manchester.ac.uk (A.K.P.J.); christopher.brown@liverpool.ac.uk (C.A.B.); 3Department of Psychological Sciences, University of Liverpool, Liverpool L69 3BX, UK; 4Salford Royal NHS Foundation Trust, Department of Neurology, Manchester Academic Health Science Centre, Salford M6 8HD, UK; christopher.kobylecki@manchester.ac.uk (C.K.); monty.silverdale@manchester.ac.uk (M.A.S.); 5Advanced Center for Electrical and Electronics Engineering, Federico Santa María Technical University, Valparaíso 1680, Chile; grace.whitaker@uv.cl; 6Centro de Investigación y Desarrollo en Ingeniería en Salud, Universidad de Valparaíso, Valparaiso 1680, Chile; wael.el-deredy@uv.cl

**Keywords:** pain, EEG, dopamine, D2 receptor, cabergoline, amisulpride, source localisation, anticipation, uncertainty

## Abstract

Striatal dopamine dysfunction is associated with the altered top-down modulation of pain processing. The dopamine D2-like receptor family is a potential substrate for such effects due to its primary expression in the striatum, but evidence for this is currently lacking. Here, we investigated the effect of pharmacologically manipulating striatal dopamine D2 receptor activity on the anticipation and perception of acute pain stimuli in humans. Participants received visual cues that induced either certain or uncertain anticipation of two pain intensity levels delivered via a CO_2_ laser. Rating of the pain intensity and unpleasantness was recorded. Brain activity was recorded with EEG and analysed via source localisation to investigate neural activity during the anticipation and receipt of pain. Participants completed the experiment under three conditions, control (Sodium Chloride), D2 receptor agonist (Cabergoline), and D2 receptor antagonist (Amisulpride), in a repeated-measures, triple-crossover, double-blind study. The antagonist reduced an individuals’ ability to distinguish between low and high pain following uncertain anticipation. The EEG source localisation showed that the agonist and antagonist reduced neural activations in specific brain regions associated with the sensory integration of salient stimuli during the anticipation and receipt of pain. During anticipation, the agonist reduced activity in the right mid-temporal region and the right angular gyrus, whilst the antagonist reduced activity within the right postcentral, right mid-temporal, and right inferior parietal regions. In comparison to control, the antagonist reduced activity within the insula during the receipt of pain, a key structure involved in the integration of the sensory and affective aspects of pain. Pain sensitivity and unpleasantness were not changed by D2R modulation. Our results support the notion that D2 receptor neurotransmission has a role in the top-down modulation of pain.

## 1. Introduction

Impaired dopaminergic transmission has been associated with chronic pain conditions [1,2,3] such as Fibromyalgia [4,5], atypical facial pain [6], and pain in Parkinson’s disease [7,8]. However, dopamine is unlikely to have a direct role in coding the intensity of pain; the majority of dopaminergic neurons do not respond directly to aversive stimuli, with only a small percentage (5–15%) reportedly being activated during pain perception [9]. Instead, changes in the dopaminergic system have been associated with the chronification of pain [10,11,12].

Dopamine has a widespread modulatory role in a variety of cognitive and affective processes such as attention flexibility [13,14,15], reward processing [16], salience assignment [17,18,19,20,21], motivational states [22,23], and emotional processing [24,25,26]. In terms of pain processing, recent theories of dopamine have focussed on its potential top-down modulatory function by considering its role within predictive coding, in which dopamine is hypothesised to modulate the precision (reliability) and confidence of subjective predictions about pain [27,28], such as those occurring during the anticipation of a painful event [29]. 

On this basis, one possibility is that the analgesic qualities of dopamine are related to the top-down cognitive and affective modulation of pain rather than modulation of afferent nociception [30,31,32]. In support of this idea, dopamine appears to interact with systems involved with the endogenous (top-down) modulation of pain [33,34,35]. Propositions of dopamine function in modulating pain include processing stimulus salience (i.e., the extent to which pain stands out and captures attention) [17,36,37,38,39], the assignment of emotional valence [40], and the modulation of sensorimotor and motivation-related processes [41,42]. However, there is currently a lack of evidence in support of dopamine modulating the anticipation of pain.

A possible molecular substrate for top-down modulation of pain anticipation is the population of the dopamine D2-like receptor (D2R) family within the striatum, which has been associated with the modulation of pain [43,44]. The D2R family consists of the D_2_, D_3_, and D_4_ dopaminergic receptors. For instance, direct injection of the D2R agonists apomorphine or quinpirole into the striatum reduced pain-related behaviours in rodents, whilst a D1-like receptor (D1R) agonist did not [43,45]. In humans, PET imaging has shown that striatal D2R binding potential is inversely correlated with an individuals’ pain tolerance of the cold pressor test [44] and heat thermode [46], as well as their ability to suppress pain [44]. However, the role of the D2R in the top-down modulation of pain processing has yet to be directly investigated. 

Top-down modulation has been researched via recording anticipation-evoked neural processing prior to the receipt of pain [29,47,48,49]. During the anticipation of a painful stimulus, the brain incorporates information from the environment and recalls prior experiences to construct a mental representation of the forthcoming pain. The brain also incorporates cognitive factors such as the current emotional state. During the anticipation period, these factors modulate the perception of the pain [50,51]. The brain regions which have been shown to be active during the anticipation of pain include the prefrontal cortex, the cingulate cortex, and the insula [52,53,54,55], all of which have connections with limbic regions, and together incorporate the subjective perception of pain [12,56].

D2R receptors are associated with non-nociceptive anticipatory processing within the prefrontal cortex [57], cingulate cortex [58,59], insula [60], and somatosensory thalamocortical connections [61]. The D2R’s role in nociceptive anticipation is yet to be reported. Therefore, this study uses EEG with source localisation and behavioural metrics (pain rating, pain unpleasantness and pain sensitivity) to compare the effects of a D2R agonist and antagonist to a control condition. We investigate the modulation of the D2R neural activations during the anticipation and the receipt of pain. The research was conducted in a healthy cohort with the aim to better understand how altered D2R processing may play a part in chronic pain conditions. 

## 2. Materials and Methods

### 2.1. Ethical Approval

The study was approved by the University of Manchester Ethics Committee and by the UK Health Research Authority for the use of a National Health Service (NHS) site (Salford Royal NHS Foundation Trust Hospital).

### 2.2. Drug Selection

We selected a D2R agonist, *Cabergoline*, and a D2R antagonist, *Amisulpride*, which both have a high affinity for D2Rs.

*Cabergoline* is a potent agonist of dopamine D2Rs, with high affinity of D_2_ (IC_50_ = 3.0 nM/L, K_i_ = 0.69 nM) and D_3_ (IC_50_ = 4.0 nM/L, K_i_ = 1.5 nM) receptors within the striatum [62,63,64], and typically used to increase dopamine in people with Parkinson’s. Cabergoline has a higher affinity for D2Rs than other agonists such as Pergolide and Bromocriptine [62]. *Cabergoline* has low affinity for D1, serotonergic (5-HT_1D_, 5-HT_2A,_ 5-HT_2B_), and adrenergic receptors (α_1_ and α_2_) [62]. The affinity for these receptors is negligible compared to the high affinity at D2Rs and will therefore not be discussed in the interpretation of the results

Amisulpride is a D2R antagonist selective for D_2_ and D_3_ receptor subtypes and typically used as an anti-psychotic. In an in vivo rat model, Amisulpride was shown to inhibit D2R binding within the striatum and limbic system, with preferential action within the limbic system [65]. Amisulpride has a high affinity and selectivity for D_2_ (K_i_ = 2.8 nM) and D_3_ (K_i_ = 3.2 nM) [66] and moderate affinity to serotonergic receptors 5-HT_2B_ (K_i_ = 13.0 nM) and 5-HT_7_ (K_i_ = 11.5 nM) (5-HT_7A_ is the most commonly expressed variant of the 5-HT_7_ receptors) [67]. D_2_R/D_3_R affinity predominates over 5-HT_7_ receptor affinity in a ratio of ~ 3:1 [67]. The 5-HT_2B_ receptors have not been directly associated with pain processing [68], [69], whereas the 5HT_7A_ receptor is involved in pain processing—primarily at the spinal level [70,71,72]. Both receptors will be considered in the interpretation of our results.

Previous studies have shown that at low doses of D2R drugs, the binding of the presynaptic D2 auto-receptor is favoured and produces the opposite of the desired results (i.e., agonist decreases dopaminergic signalling and vice versa) [73,74] (see Appendix A, Figure A1). Therefore, a sufficiently high dose was selected to modulate the postsynaptic D2R and resulted in the agonist increasing dopaminergic D2R signalling and the antagonist reducing D2R signalling. The dosage for *Cabergoline* (1.25 mg) and *Amisulpride* (400 mg) was based on previous studies which demonstrated cognitive effects in the appropriate direction at the chosen dose with minimal side effects (Cabergoline: [75,76] and Amisulpride: [77]). Cabergoline has an elimination half-life of between 63 and 69 h [78], and peak plasma concentrations are observed between 2 and 3 h [78]. Amisulpride has an elimination half-life of 12-h after an oral doseand peak plasma concentrations are observed at approximately 3 h [79].

### 2.3. Participant Recruitment

A total of 29 healthy participants (16 females) were recruited for the study (mean age 22.4 years, SD 3.2 years). All subjects gave written informed consent. The exclusion criteria of the study were as follows: history of significant head injury or seizures, diagnosed or taking medication for any neurological or psychiatric condition, history of drug or alcohol dependence, use of psychotropic medication within the past 6 months, use of dopaminergic drugs within the past month or lifetime use exceeding 3 months, pregnant or breastfeeding or attempting to conceive, and suffering from chronic pain. The phase of menstrual cycle was not an exclusion criterion and was not recorded at experimental visits.

To limit the variability in baseline dopamine level between subjects, prior to recruitment, all participants completed the Barratt Impulsivity Scale (BIS-11, [80]) subscale of ‘cognitive instability’. The degree of impulsivity has been shown via positron emission tomography (PET) and single-photon emission computed tomography (SPECT) imaging studies to be significantly correlated to dopaminergic transmission and receptor/transporter abundance within the striatum [81,82,83,84]. Participants were only recruited if their score was within 1.5 standard deviations of the mean reported from a large control sample (1577 healthy adults) [85]. It is important to note that no imaging was carried out within our participants to confirm our use of the BIS-11 to limit variability in baseline dopamine. 

The study used a repeated-measures, double-blind, and triple-crossover design such that all participants were recruited to attend three visits and complete the experimental protocol following ingestion of the agonist, antagonist, and control substance. Two participants did not complete all three visits of the study due to withdrawing from the study, and one participant was removed due to an adverse reaction to Amisulpride. Therefore, 26 datasets were included for behavioural and EEG analyses (mean age 22.2 years, SD 3.7 years, 14 females). 

### 2.4. Health Screening

Prior to the first experimental visit, the participants attended a health screening to deem them safe to take part in the study. A medical doctor assessed the participants’ heart rate, blood pressure, and temperature and recorded an electrocardiogram (ECG). Due to the potential risk of Amisulpride causing arrhythmias, all participants were assessed for QTc abnormalities in the ECG prior to inclusion in the study. No participant presented with any abnormalities resulting in the exclusion from the study. A letter to the participant’s general practitioner (GP) was also organised to inform them of the participant’s involvement in the study. 

### 2.5. Prior to Experimental Visits

Participants were instructed to not consume alcohol for 24 h prior to the visit, to only drink their normal intake of coffee or tea on the morning of each visit, and to refrain from consuming other caffeinated drinks 2 h prior to each visit. Participants were also informed to not consume any psychoactive substances for the duration of the study or any over-the-counter medicine 48 h prior to each visit. 

### 2.6. Experimental Visit Protocol

#### 2.6.1. Drug Administration

One of three substances was administered orally to the participant at each experimental visit: Cabergoline (Agonist) (1.25 mg), Amisulpride (Antagonist) (400 mg), or Sodium Chloride (Control) (20 mg). All drugs hold a full product licence (EU). All participants were provided with the potential side effects of the drugs. Participants were instructed that the drugs may or may not modulate dopamine levels, and no further details were provided regarding the potential effect on pain experience. The administration of the drug was double-blinded, and each participant had each drug condition once over the three visits in a randomised order. The 3 visits were separated by at least 10 days. The administration of the drug was recorded as time 0 h, and experimental testing commenced at 3 h 30 min post drug administration. 

#### 2.6.2. Eye Blink Rate

Previous research has indicated a potential correlation between the level of striatal dopamine and the rate of blinks per minute [86,87]. The blink rates of the participants were recorded using frontal EEG electrodes during a resting state of 9 minutes at +2 h after taking the drug and prior to any experimental testing. The participants were not informed that their blink rate was being assessed to avoid affecting their spontaneous blink rate. 

#### 2.6.3. Pain Stimuli

A CO₂ laser [50 W Synrad 48-5 J-series (J-48-5(S)W) Wavelength: 10,600 nm] was used to deliver acute pain to the right dorsal forearm surface [53]. The CO_2_ laser activates the A-delta and C-fibres at conduction velocities of ~10 m/s and ~1.0 m/s, respectively, producing a localized ‘sharp pricking’ sensation and ‘diffuse’ and ‘burning’ after-sensation [88,89]. The acute and fast activation of the A-delta fibres allowed for a precise anticipation period. In addition, the CO_2_ laser has been used to investigate the anticipatory response in previous studies [51,90,91,92]. The CO₂ laser delivered a beam with a diameter of 15 mm and 150 ms duration. The laser intensity was measured in voltage (V) and ranged from 0.6 V to 2.6 V. For each test, the stimuli were delivered in an area measuring 4 × 5 cm and were delivered in a predetermined randomised path [51]. This was to avoid habituation, sensitization, or skin damage.

#### 2.6.4. Psychophysics

Before starting the experimental protocol, psychophysics was used to calibrate the laser to the individual’s pain sensitivity. An ascending method of limits procedure was used, starting from 0.6 V, with 0.06 increments each time. The participant used an 11-point numerical rating scale (NRS) (0−10) to rate the intensity of the pain perceived for each laser stimuli (0 = no sensation, 4 = pain threshold, 7 = moderately painful, 10 = unbearably painful). The rating scale was introduced to the participant via these standardised descriptives to ensure that no explanation altered their interpretation of the scale. The procedure was repeated three times to allow participants to become accustomed to the laser. The average voltage to induce level 4 (low) and level 7 (moderate) pain were used to provide ‘low’ and ‘high’ pain stimuli, respectively, for the main laser experiment protocol.

#### 2.6.5. Main Experiment

The participants received 120 laser stimuli at the two intensities (low and high) separated into four conditions: Certain Low (level 4), Certain High (level 7), Uncertain Low (level 4), and Uncertain High (level 7). To investigate the anticipation to a painful stimulus, a 3-s auditory countdown preceded the laser stimuli (see Figure 1). The first auditory cue was presented concurrently with a visual anticipatory cue to indicate the forthcoming laser stimuli. The participant was either shown the word: ‘Low’, ‘High’ (certain anticipation), or ‘Unknown’ (uncertain anticipation). The presentation of the word ‘Unknown’ indicated that the laser stimulus has an equal chance of being low or high intensity. This was to investigate the importance of the degree of certainty in the anticipation of the laser stimuli. The image was also used as a visual fixation cue to discourage eye movements. After each laser stimulus, the 0–10 NRS was shown on the screen, and the participant rated the intensity of the pain via a numerical keypad. The order of the stimuli was randomised and separated into three experimental blocks of 40 stimuli, with 2-min breaks in-between. Following each block, the participants rated the unpleasantness on average for each laser condition *(Certain Low, Certain High, Uncertain Low, and Uncertain High).* Unpleasantness was scored using an 11-point NRS, whereby 0 is not unpleasant and 10 is the most unpleasant sensation. 

#### 2.6.6. EEG Recording

A BrainVision MR EEG cap was used to record data from 63 scalp electrodes using a BrainVision-cap system [Standard BrainCap-MR with Multitrodes]. The arrangement of the electrodes was modelled on the extended 10–20 system. Recording parameters were set at Filter (DC to 70 Hz), Sampling rate (1000 Hz), and Gain (500). To reduce electrical interference, a 50 Hz notch filter was applied. Prior to starting the laser protocol, resting states were recorded with eyes open and closed for two minutes in all participants. This ensured that the experience prior to the experiment was identical. 

### 2.7. Analysis Methods

#### 2.7.1. Statistical Analysis of Behavioural Data

Statistical analyses of the behavioural measures were carried out using IBM SPSS Statistics 22 software. Prior to using statistical tests, the data were assessed for normality using a combination of Q–Q plots, histograms, and the values of skew and kurtosis. To assess whether dopamine modulation evoked changes in sensitivity to the laser stimuli, the participants’ tolerance to the laser (level 7 voltage) was compared between each drug condition using a repeated measures one-way ANOVA. In addition, the pain and unpleasantness rating of the experimental laser stimuli were investigated for drug-related changes via a repeated measures three-way ANOVA with within subject factors of drug (*control, agonist, and antagonist*), certainty (*certain and uncertain*), and intensity (*low and high*).

#### 2.7.2. EEG Analysis Method

EEG pre-processing was carried out using EEGLAB toolbox [93] in MATLAB version R2015a (The Mathworks Inc), whilst statistical analysis was carried out using Statistical Parametric Mapping (SPM12) toolbox (Wellcome Department of Imaging Neuroscience, Institute of Neurology, UCL, London, UK) running in MATLAB. 

#### 2.7.3. Blink Rate

For each participant and visit, the number of blinks per minute was calculated using the frontal EEG electrodes of the resting state recording. The data were downsampled to 200 Hz, and eye blinks were identified using Independent Component Analysis (ICA) within the EEGLAB toolbox. ICA is used to identify and remove ocular artefacts, by discarding the components that contain them. A Matlab script used those eye blink dominant components, identified by its topography and morphology, to count the number of blinks in a stretch of 9 min of resting EEG recorded before the start of the main study. The blinks per minute were reported as the blink rate and were assessed for differences between dopamine manipulation and blink rate via contrast analysis. 

#### 2.7.4. EEGLAB Pre-Processing

Pre-processing consisted of the removal and interpolation of bad channels, down-sampling to 500 Hz, a low-pass filter of 20 Hz, and re-reference to the common average. The four conditions were separated, −3500 ms to 2000 ms epochs were extracted centred on the delivery of the laser stimulus, and Linear detrend was applied. Independent Component Analysis (ICA) was carried out on all datasets using the SemiAutomatic Selection of Independent Components for Artifact correction (SASICA) toolbox to select components to remove via pre-determined thresholds. The thresholds were set to Autocorrelation (threshold = 0.35 r, lag = 20 ms), Focal (threshold = 3.5 z), Focal trial (threshold 5.5 z), Signal-to-noise (period of interest (POI) = [0 Inf], baseline (BL) [-Inf 0], threshold ratio = 0.5), and Adjust Selection enabled. The thresholds were sufficient to remove artefacts from the majority of the datasets; however, a number of datasets required further manual removal of eye-blink components were not identified by SASICA. 2.7.5. SPM EEG Analysis

The pre-processed datasets were converted to SPM-compatible files. Statistical analysis was carried out to investigate the anticipation-evoked potentials and the post-stimulus laser evoked potentials (LEPs) using scalp-level and source localisation analysis techniques available in the SPM toolbox. 

SPM scripts for batch processing were used to analyse the EEG data at the scalp level and for source localisation. Two baselining methods were applied to the data analysis for the anticipation phase and were applied for scalp-level and source localisation analysis. Primary analysis applied distinct baselines (BLs) of 500 ms, occurring prior to each of the three auditory cues to analyse each of the three anticipation phases (early, mid, and late) [94]. The BLs and time window of interest (TWOI) were as follows: Early [BL: −3500 ms −3000 ms: TWOI: −2500 ms −2000 ms], Mid [BL: −2500 ms −2000 ms: TWOI: −1500 −1000 ms], and Late [BL: −1500 −1000 ms: TWOI: −500 0 ms] anticipation phases. The aim of baselining uniquely for each anticipation window was to reduce variability in the data as the anticipation phase progressed, such that the analysis of each phase of anticipation was unique to that phase and not subject to variability arising from neural activity occurring in the previous phase. The secondary analysis method applied a single baseline of 500 ms prior to the first auditory cue [BL: −3500 ms −3000 ms] that was common to every TWOI anticipation phase (early, mid, and late). This second analysis was conducted to enable comparison to previous studies [51,53] that used the same baselining method and to explore to what extent the results from the primary analysis were dependent on the baselining method used. All statistical analysis for the anticipation phase was adjusted for multiple comparisons. 

Analysis for the post-stimulus phase TWOI [200 ms 600 ms] (centred on the LEP), was baseline corrected to −500 ms prior to the laser stimulus [BL: −500 ms 0 ms]. 

#### 2.7.5. Source Localisation Analysis Parameters

SPM12 EEG and MATLAB scripts were used to estimate the sources of the anticipation- and laser-evoked potentials using Low-Resolution Electromagnetic Tomography (LORETA). The forward model was created using an 8196 vertex template cortical mesh coregistered to the electrode positions of the standard 10–20 EEG system. A three-shell boundary element (BEM) EEG head model available in SPM12 was used to compute the forward model. The images were smoothed with a 12 mm full-width-at-half-maximum (FWHM).

Source localisation results were reported as follows: To control for multiple comparisons, a cluster-forming threshold of *p* < 0.001 was used, and resulting clusters were considered significant at FWE (*p* < 0.05). Significant clusters were also restricted to >100 voxels in size and regions labelled using the Anatomical Automatic Labelling (AAL2) toolbox in SPM. To investigate potential changes in all brain structures, initial analyses were not restricted to regions of interest. Follow-up small volume correction (SVC) was applied using a 25 mm sphere to further investigate effects within regions of the pain matrix, specifically the Insula, Anterior Cingulate Cortex (ACC), Thalamus, and Amygdala, using coordinates from the Brede Database [95]. Significant results following SVC are highlighted in the results tables.

#### 2.7.6. EEG Analysis Statistical Analysis

The anticipatory time windows of interest (TWOIs) were analysed using a repeated-measure three-way ANOVA, with within-subject factors of Drug *(Control* vs. *Agonist* vs. *Antagonist),* Certainty *(Certain* vs. *Uncertain),* and Expectation *(Certain Low* vs. *Certain High).* The analysis of the post-stimuli TWOI was conducted using a repeated measure three-way ANOVA, with within subject factors of Drug *(Control* vs. *Agonist* vs. *Antagonist)*, Certainty *(Certain* vs. *Uncertain*), and Intensity *(Low* vs. *High).*

## 3. Results

### 3.1. Behavioural Results

To assess whether dopamine modulation evoked changes in pain sensitivity, the participants’ tolerance to the laser was compared between each drug condition. The mean ± SD laser voltage to induce high (level 7) pain was calculated for control (1.93 ± 0.43 V), agonist (1.98 ± 0.49 V), and antagonist (1.97 ± 0.41 V). There was no significant drug-induced effect on sensitivity to the nociception F(2, 50) = 0.465, *p* = 0.631, (Table 1).

In addition, the mean pain rating score for each drug condition was calculated. The pain rating scores were normally distributed, as assessed by Shapiro–Wilk’s test of normality (*p* > 0.05). The within-subject comparison of the three Drug Conditions (Control, Agonist, and Antagonist) reported no effect on the pain rating F(2, 50) = 0.60, *p* = 0.552. This outcome was expected, as the laser intensity was calibrated to the participants’ individual pain sensitivity on each visit to induce a score of seven on the pain NRS.

As expected, there was a significant effect of intensity [F(1, 25) = 262.98, *p*< 0.001], such that high-intensity laser stimuli were rated as more painful than low intensity. The effect of Certainty (*Certain* vs. *Uncertain*) on the pain rating did not show a significant result F(1, 25)= 3.33, *p* = 0.80; however, a two-way interaction between Certainty and Intensity was shown [F(2, 50) = 47.82, *p* < 0.001], such that pain ratings increased from Certain Low (2.95 ± 0.85) and Uncertain Low (3.36 ± 0.98), but decreased between Certain High (6.31 ± 0.60) to Uncertain High (5.73 ± 0.56) (i.e., Pain rating: *Certain Low < Uncertain Low < Uncertain High < Certain High*). Therefore, the ‘Uncertain’ cue results in a reduced ability to distinguish between low and high laser stimuli.

Interestingly, there was a three-way interaction between Drug, Certainty, and Intensity [F(2, 50) = 5.60, *p* = 0.006]. This indicates that the modulation of D2Rs affected the interaction between Certainty and Intensity. Follow-up three-way ANOVAs established how the modulation of D2R compared to control (control vs. agonist, and control vs. antagonist) affected the interaction between certainty and intensity. Firstly, the comparison of control and agonist revealed no three-way interaction between Drug, Certainty, and Intensity [F(1,25) = 2.658, *p* = 0.116]. However, the comparison of control versus the antagonist revealed a significant three-way interaction, [F(1, 25) = 9.535, *p* < 0.005]. The effect of the uncertain cue increasing Low pain rating, and decreasing High pain rating was larger in the antagonist condition compared to the control condition. Hence, the ability to distinguish between low and high stimuli following the ‘Unknown’ cue was reduced in the antagonist condition.

There was no significant difference in the mean unpleasantness rating for each drug condition (Control, Agonist, and Antagonist) [F(2, 30) = 2.766, *p* = 0.079] (Table 2). There was an effect of Intensity such that high-intensity stimuli were rated more unpleasant than low-intensity stimuli [F(1, 15) = 197.75, *p* = 0.000]. There was also an effect of Certainty (Certain vs. Uncertain) [F(1, 15) = 11.924, *p* = 0.004] on unpleasantness ratings such that the uncertain cue of “Unknown” caused a higher rating in unpleasantness compared to the Certain cues of “Low” and “High”. This effect was consistent across all three drug conditions.

### 3.2. EEG Results

#### 3.2.1. Blink Rate Analysis

The blink rate assessed via EEG was shown to have a significant linear relationship between striatal dopamine activity (F(1,27) = 7.242, *p* < 0.05, ηp^2^ = 0.211; Figure 2). The agonist increased the eye-blink rate, whilst the antagonist reduced the eye-blink rate, in comparison to the control condition, which is consistent with previous reports [96,97,98,99].

#### 3.2.2. Scalp-Level EEG Analysis

The electrode-level analysis of the anticipation TWOIs (early, mid, and late) reported no main effect of Drug, Expectation, or Certainty for both baselining methods. 

In addition, the analysis of the post-laser time window of interest (TWOI) [200 to 600 ms] at scalp-level also reported no significant effects of drug or certainty. A main effect of intensity produced two clusters (*p* < 0.001, *k*_E_ = 20,418, x: −13 mm y: −36 mm, 464 ms; and *p* < 0.000, *k*_E_ = 3510, x: 55 mm y: −68 mm, 450 ms) with the T-contrast showing an enhanced activation during high pain in contrast to low pain (*p* < 0.001, *k*_E_ = 17,732, x: −13 mm, y: −41 mm, 460 ms).

#### 3.2.3. Source EEG Analysis

##### Anticipatory Time Window of Interest

The source localisation analysis identified significant differences between conditions within the anticipatory- and laser-evoked time windows using the pre-cue baselines. Firstly, during mid-anticipation, both the agonist and antagonist reported a lower degree of activity within several regions compared to the control condition (see Table 3). The agonist induced lower activity in the right mid-temporal region and the angular gyrus. The antagonist induced reduced activity within the right postcentral, mid-temporal, and inferior parietal regions (Figure 3).

The expectation of a higher pain *(Certain High)* during mid-anticipation (Table 3) increased activation within the left mid-temporal, mid-occipital, and insula regions compared to the expectation to lower pain *(Certain Low).*

##### Post-Laser Time Window of Interest

The higher laser intensity evoked higher activation within the post-laser TWOI [200 ms to 600 ms] (High > Low: *p* < 0.001, F = 8.44, *kE* = 98,792, x: 8, y: −14, z: −16). Interestingly, there was a main effect of the drug conditions within the post-stimulus TWOI (Table 3). The antagonist induced a reduced level of activation within the right insula in comparison to the control condition (see Figure 4). In comparison to the agonist condition, the antagonist showed a lower activation within the right hippocampus, the mid- and inferior temporal regions, the insula, and Heschl’s Gyrus *(auditory processing)* (see Figure 4).

## 4. Discussion

This study investigated the role of the D2 dopaminergic receptors during pain anticipation and pain perception. To summarise, the key findings were the following: (1) The antagonist reduced an individuals’ ability to distinguish between low and high pain following uncertain anticipation, (2) the agonist and antagonist reduced neural activations in specific brain regions during anticipation and receipt of pain, and (3) pain sensitivity and rating of unpleasantness were not changed by D2R modulation.

The agonist and antagonist evoked changes in the resting blink rate—a method to measure striatal dopamine activity [96,97,98,99]. The agonist increased the blink rate, indicating higher dopamine signalling. The antagonist decreased the blink rate, indicating reduced dopaminergic signalling. Therefore, this suggests that the agonist and antagonist dosage used was sufficient to induce the desired dopamine modulation. However, there are contradictory reports of the accuracy of using the blink rate to quantify striatal dopamine activity [101,102], and therefore, without PET imaging to quantify dopaminergic activity, we cannot confirm changes in dopaminergic signalling.

### 4.1. Antagonist Effect on Pain Rating

The rating of the uncertain laser stimuli was modulated by the D2R antagonist, *amisulpride*. Across all drug conditions, uncertainty affected low and high pain differently, such that *Uncertain Low* was rated higher than *Certain Low,* whereas *Uncertain High* was rated lower than *Certain High.* Firstly this indicates that uncertainty reduces the ability to accurately rate the intensity of pain and is consistent with previous research [29,103]. The antagonist increased the effect of uncertainty on the pain rating, whereby (compared to the control) the pain rating was even higher for *Uncertain Low* and even lower for *Uncertain High,* therefore indicating that the reduction of dopaminergic signalling via D2Rs (and inhibition of 5-HT_2B_/_7A_ receptors) resulted in a further reduced ability to distinguish between low and high pain when no information was provided prior to the stimuli.

Dopamine is important in the certainty and confidence of decisions and perceptions within the environment [104,105]. Schwartenbeck et al. (2015) proposed that confidence and precision in choices are encoded by the dopaminergic activity of the midbrain [27]. The Bayesian brain hypothesis, which states that perception is the integrated outcome of prior expectation and sensory information, present a potential explanation of the results. The antagonist may increase the weight of the prior (visual cue) and reduce the precision/confidence in the sensory information (laser stimulus), whereas the agonist may reduce the weight of the prior and increase the precision/confidence of the sensory information. Although no study has investigated the accuracy of pain perception, a study by Tomassini et al. (2016) reported that D2R antagonism led to increased uncertainty and reduced precision of the representation of time [106]. Therefore, we propose that the reduced dopaminergic signalling via the antagonist resulted in increased dependence on the visual cue and, therefore, decreased confidence in the sensory perception of the low and high pain stimuli following uncertain anticipation.

The antagonist also inhibits transmission at 5-HT_2B_/_7A_ receptors at a lower affinity than D2Rs. Serotonergic transmission is documented to be involved in pain processing, particularly at the spinal level and within the descending pain pathway [107,108,109,110]. A global loss of serotonin has been associated with a reduced ability to discriminate laser stimuli intensity [111]; however, there is no evidence to show that 5-HT_2B_/_7A_ receptors are associated with pain discrimination.

### 4.2. D2R Modulation Reduced Neural Activity during Anticipation

In comparison to the control condition, the agonist reduced activity in the right mid-temporal region and the right angular gyrus, whilst the antagonist reduced activity within the right postcentral, right mid-temporal, and right inferior parietal regions. These regions are known to be innervated by striatal dopaminergic neurons [112,113] and are involved in the top-down processing of salient stimuli. Therefore, we demonstrate that these regions rely (in some part) on the action of D2R-innervation during pain processing.

The agonist and antagonist reduced activity in the right mid-temporal lobe, a region rich in D2Rs [114] involved in sensory integration and semantic processing [115,116,117,118], and has shown increased activity during uncertain (non-nociceptive) anticipation [119] and pain anticipation [91]. The agonist also reduced activity in the angular gyrus, a region located between the temporal and parietal lobes, and is involved in integrating multi-sensory information to give sense to events and reorient attention to salient or informative stimuli [120,121,122,123,124]. Kluger et al. (2017) reported how uncertainty increased activity in the right angular gyrus [119].

The antagonist reduced activity in the right postcentral gyrus and the inferior parietal region. The postcentral gyrus is known to activate during pain processing [125], whilst the inferior parietal cortex is part of the right-lateralised attention network and is associated with the intentional focusing of attention toward pain, and in particular its spatial location [126,127]. A previous study using the same protocol reported increased activation of the inferior parietal cortex during anticipation [91]. This study demonstrates that these processes may involve D2R signalling during the top-down processing prior to pain receipt.

A further important consideration is the antagonist’s additional inhibition of 5-HT_2B/7A_ receptors (albeit with lower affinity). To the best of our knowledge, there is no evidence that the 5-HT_2B/7A_ receptors are involved in anticipatory processing and sensory integration. Nevertheless, 5-HT_7_ receptors are present within the mid-temporal lobe [128] and may have contributed to the reduced activity during anticipation seen in this study.

It is important to note that the agonist and antagonist both reduce neural activity despite having the desired opposite effect on dopamine signalling. One interpretation we propose is that the dose of the agonist or antagonist was not sufficiently high enough to increase or decrease dopaminergic transmission, respectively (due to the action of presynaptic D2R auto-receptors, see Figure A1), and instead both conditions result in a decrease/increase in dopamine signalling. However, the dosage selected consistent with previous reports indicating a desired pharmacodynamic effect and the resting blink-rate (a marker of striatal dopamine activity) reported a successful increase in dopamine in the agonist and a decrease in the antagonist condition.

Therefore, an alternative interpretation could be due to the inverted-u relationship of dopaminergic responses, such that both low and high dopamine levels produce the same output [126,127,128,129,130,131,132,133,134,135,136]. For instance, research by Friston et al. (2012) using Bayesian simulations demonstrated how “*a single functional role for dopamine at the synaptic level can manifest in different ways at the behavioural level*” [137]. However, the inverted-u relationship of dopamine is most commonly discussed in terms of behavioural outcomes, rather than neural activity, and therefore may be too simplistic to explain the current results.

### 4.3. Antagonist Reduced Neural Activity during Receipt of Pain

During receipt of pain, the antagonist induced a reduced level of activation within the right insula in comparison to the control condition, and reduced activity in the right hippocampus, mid- and inferior temporal region, and insula, when compared to the agonist condition.

The insula has been considered to be involved in coding intensity of pain [138] but likely plays a more complex role in integrating the sensory and affective aspects of pain [139,140]. The insula integrates top-down expectations with bottom-up nociceptive information [51,140,141]. Hence, D2R antagonism resulting in a reduced activation of the insula may reflect deficits in the optimal integration of sensory, affective, and cognitive information. This deficit in the insula’s integration of information may manifest behaviourally in the altered rating of the uncertain pain stimuli.

Dopamine’s role in the perception of stimuli has been investigated via the oddball task, whereby dopamine response increases when an unexpected tone is presented. In a rodent study, D2R antagonist (*Eticlopride*) reduced the neural activity within the midbrain during receipt of unexpected tones in the oddball paradigm [140], thus demonstrating how D2R antagonism can reduce neural activity in response to salient stimuli.

### 4.4. No Change in Pain Sensitivity or Unpleasantness

The individual’s pain sensitivity and rating of unpleasantness were not changed by D2R modulation. A study by Becker et al. (2013) reported similar results, such that neither Sulpiride (D2R (D_2_ and D_3_) antagonist) nor dopamine precursor depletion altered the unpleasantness ratings of thermal pain [141]. However, there are contrasting results from Tiemann et al. (2014), whereby they reported that an acute depletion in the dopamine precursor increased pain unpleasantness (not pain sensitivity) to laser stimuli [30]. Differing pain-delivery and dopamine manipulation methods and the acute action of modulation are likely to explain the differing findings.

### 4.5. Clinical Impact

Clinical research questions that follow from this study include whether chronic dopamine dysfunction (as found in conditions such as Parkinson’s disease and Fibromyalgia) might impair the top-down modulation of pain, resulting in chronic pain. The acute administration of D2R agonist and antagonist drugs may not be sufficient to understand how long-term changes in dopaminergic transmission can impact pain processing. However, the significant differences we have found following acute D2R manipulation within aspects of pain processing, along with previous research highlighting D2R dysfunction in chronic pain conditions [6], suggests that prolonged changes in dopamine signalling might have profound effects.

## 5. Conclusions

The study highlights the involvement of the dopaminergic system in top-down pain processing, and its involvement in sensory integration and accurate perception, rather than pain sensitivity. Neither drug modulated pain sensitivity or unpleasantness ratings; however, the antagonist *(amisulpride)* reduced the ability to distinguish pain intensity. EEG source localisation identified that both drugs reduced anticipatory activity in regions associated with salient stimuli processing and sensory integration. In comparison to the control condition, the antagonist reduced activity in the insula during the receipt of pain —a region associated with the integration of the sensory and affective aspects of pain. In conclusion, an acute change in D2R activation was sufficient to alter pain processing in a healthy cohort and presents a scientific rationale to investigate how chronic dysfunction of the D2R in chronic pain conditions contributes to symptoms.

## Figures and Tables

**Figure 1 brainsci-12-00351-f001:**
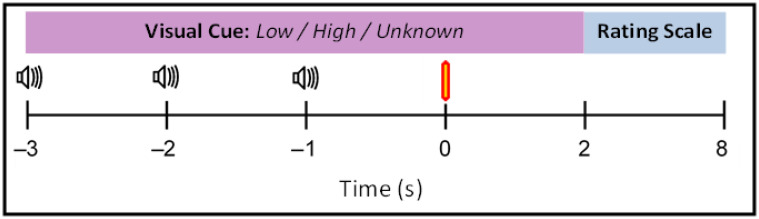
A schematic diagram of a single trial of the experimental paradigm. A computer monitor showed the participant a visual cue of “Low”, “High” or “Unknown” from −3 s to +2 s. A 3-second countdown of auditory cues at −3, −2 and −1 allowed accurate anticipation of the laser stimuli at time 0s (red bar). The presentation of the visual cue at −3 s was concurrent with the first auditory cue. At +2 s, an 11-point numerical rating scale (NRS) was presented for the participant to rate the laser stimulus.

**Figure 2 brainsci-12-00351-f002:**
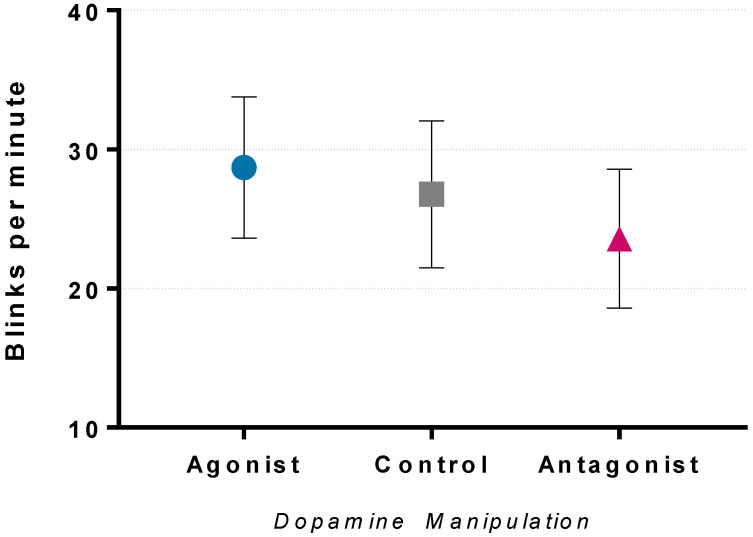
The eye-blink rate (EBR) calculated for each dopamine manipulation condition produced a significant linear relationship between striatal dopamine activity and EBR (*p* < 0.05). The D2 receptor agonist increased the EBR and the antagonist reduced the EBR in comparison to the control condition. Data are presented as mean and 95% confidence intervals (cf. [100]).

**Figure 3 brainsci-12-00351-f003:**
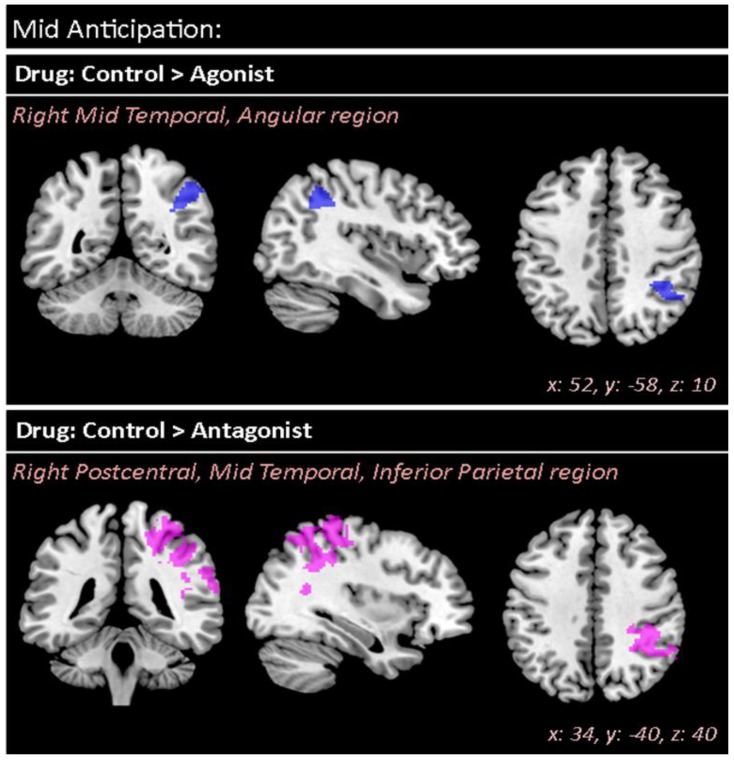
Source estimates during mid-anticipation for Control versus agonist (Cabergoline) and antagonist (Amisulpride). The time window of mid anticipation [−1500–1000 ms] was baseline corrected to [−2500–2000 ms]. Clusters are shown at FWE correction and regions labelled using the AAL2 atlas. The agonist (blue) reduced activity within the right mid-temporal and angular regions in contrast to the control. The antagonist (pink) reduced activity within the right postcentral, mid-temporal, and inferior parietal regions. The xyz position of the slices are aligned with the peak-voxel at peak-level inference and reported in mm according to the MNI atlas. In comparison to the control, the agonist reduced activity within the right mid temporal and angular regions, and the antagonist reduced activity within the right postcentral, mid temporal, and inferior parietal region.

**Figure 4 brainsci-12-00351-f004:**
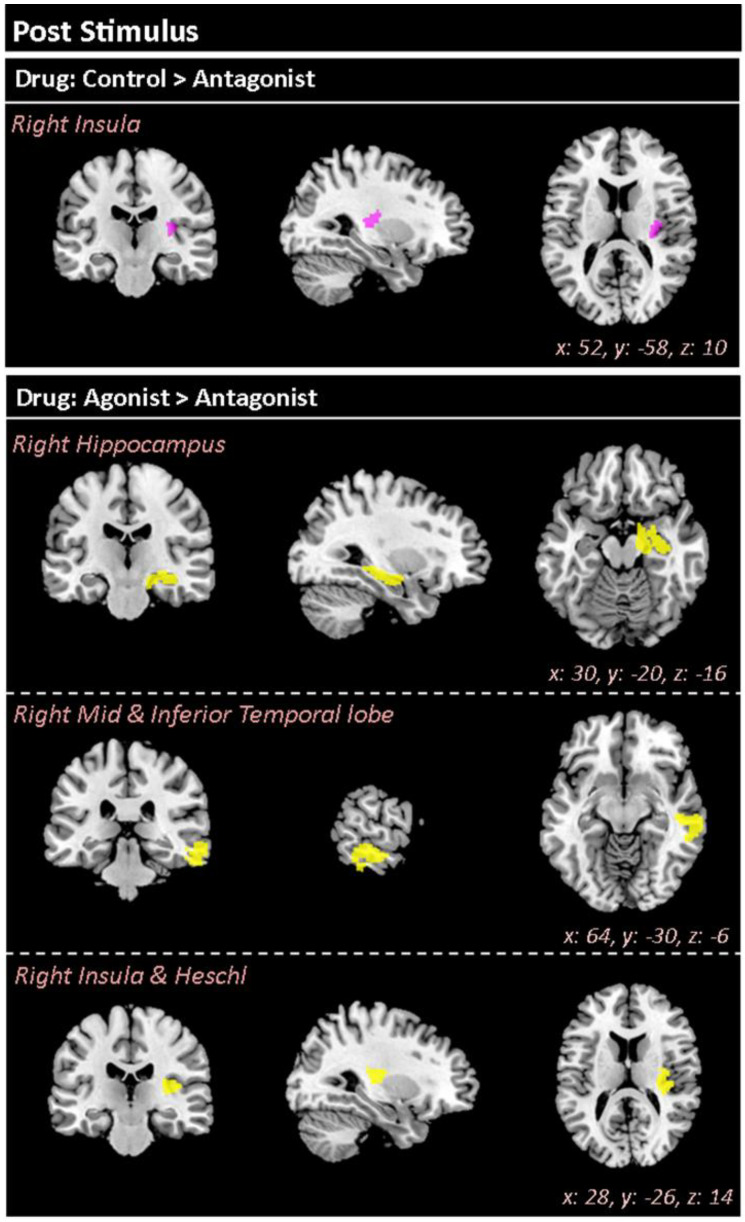
The drug effect reported using source localisation for post-stimulus TWOI [200 600 ms]. The TWOI was baseline corrected to −500 ms prior to the noxious stimuli. There was a significant difference between the control and antagonist conditions, showing a lower degree of activity within the right insula in the antagonist condition. The antagonist condition also reported a lower activity in comparison to the agonist within the right hippocampus, mid/inferior temporal lobe, insula, and Heschl’s gyrus.

**Table 1 brainsci-12-00351-t001:** Pain rating descriptive statistics. The ratings are separated into cue type and drug condition. Ratings are reported as mean and standard deviation.

Pain Rating (NRS Score/10)								
	Low		High		Uncertain Low		Uncertain High	
Drug Condition	Mean	SD	Mean	SD	Mean	SD	Mean	SD
Control	2.97	0.88	6.30	0.71	3.28	1.07	5.86	0.62
Agonist	3.05	0.93	6.28	0.79	3.50	1.14	5.75	0.80
Antagonist	2.83	1.00	6.36	0.60	3.31	1.10	5.58	0.68

NB: NRS: Numerical rating scale, SD: Standard deviation.

**Table 2 brainsci-12-00351-t002:** Unpleasantness rating descriptive statistics. The ratings are separated into cue type and drug condition. Ratings are reported as mean and standard deviation.

	Unpleasantness Rating (NRS Score/10)							
	Low		High		Uncertain Low		Uncertain High	
Drug Condition	Mean	SD	Mean	SD	Mean	SD	Mean	SD
Control	2.39	1.30	6.49	0.91	2.56	1.44	7.08	1.30
Agonist	2.25	1.07	6.49	0.67	2.35	1.02	6.98	0.90
Antagonist	1.94	1.22	5.42	2.37	2.22	1.55	5.91	2.67

NB: NRS: Numerical rating scale, SD: Standard deviation.

**Table 3 brainsci-12-00351-t003:** Significant results of the source localisation analysis within mid-anticipation and the post-stimulus TWOI. The mid-anticipation TWOI is baseline-corrected to the pre-auditory cue time window. The significant clusters were restricted to >100 voxels, reported at FWE correction with a threshold of *p* < 0.025 to account for multiple comparisons. Results are divided into main F-contrast and post-hoc T-contrasts. The MNI coordinates are reported for the peak voxel within the cluster. The brain regions are labelled using the Anatomical Automatic Labelling (AAL2) atlas, with the percentage overlap of the significant cluster with the brain region reported. The region with the highest percentage overlap is shown, unless an equivalent share of percentage overlap was observed. Brain regions labelled by the atlas as ‘Unknown’ are not shown.

		Cluster-Level		Peak-Level		MNI Coordinates			Brain Region	
		*p* _(FWE)_	*K*	F/T	Z	X	Y	Z		
**Mid Anticipation**	**F-Contrast**									
	Drug	0.004	909	11.62	4.20	32	−44	34	R	Inferior Parietal (48.0%)
		0.013	700	9.79	3.79	34	−54	14	R	Mid Temporal (56.3%)
	Expectation	0.000	4098	25.90	4.85	−44	−48	6	L L	Mid Temporal (35.3%) Mid Occipital (23.1%)
		0.013	783	18.64	4.09	−56	2	−16	L	Mid Temporal (69.1%)
		0.007	931	15.28	3.69	−14	−86	2	L L	Calcarine (52.8%) Lingual (37.2%)
		0.005	390	15.65	3.73	−30	−18	8	L	Insula (60.7%)
	**T-Contrast**									
	Drug Control > Agonist	0.031	693	3.92	3.87	52	−58	10	R R	Mid Temporal (49.9%) Angular gyrus (29.4%)
	Drug Control > Antag	0.000	4184	4.60	4.52	34	−40	40	R R R	Postcentral (17.2%) Mid Temporal (16.9%) Inferior Parietal (15.0%)
	Expectation High > Low	0.000	8555	5.09	4.98	−44	−48	6	L L	Mid Temporal (30.4%) Mid Occipital (17.9%)
		0.003^◊^	641	3.96	3.90	−30	−18	8	L	Insula (58.8%)
**Post-Stimulus**	**F-Contrast**									
	Drug	0.022	164	10.82	4.02	28	−26	14	R	Insula (40.9%)
	**T-Contrast**									
	Drug Control > Antag	0.033^◊^	131	3.73	3.68	28	−26	14	R	Insula (45.0%)
	Drug Agonist > Antag	0.022	748	3.93	3.88	30	−20	−16	R	Hippocampus (54.3%)
		0.010	940	3.80	3.75	64	−30	−6	R R	Mid Temporal (52.0%) Inferior Temporal (38.7%)
		0.008 ^◊^	347	4.14	4.07	28	−26	14	R R	Insula (32.6%) Heschl (18.9%)

NB: ^◊^ Small volume correction (SVC) in regions of interest was applied using a sphere with 25 mm radius. ‘>’ Symbolises that the left-sided condition showed a greater response in comparison to the right-sided condition. FWE = Family-Wise Error; K = number of voxels; F/T = F-value/T-value; Z = Z-score; Expectation = ‘Low’ vs. ‘High’ visual cues; Antag = Antagonist; R = Right hemisphere; L = Left hemisphere.

## Data Availability

The data presented in this study are available on request from the corresponding author. The data are not publicly available due to additional analyses being conducted on the dataset.

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
