# Peer review of "Altered Pain Processing Associated with Administration of Dopamine Agonist and Antagonist in Healthy Volunteers"

_brainsci, 2022, doi:10.3390/brainsci12030351_

Round 1

Reviewer 1 Report

The manuscript by Martin et al. deals with the effects of cabergoline and amisulpride on pain processing in humans. While the experimental design on pain processing is sound, there are some points that should be clarified.

I have following comments:

Major:

1) The title: should be changed - see below.

2) Both cabergoline and amisulpride are not D2 selective. They, at least, bind with similar affinity to D3 dopamine receptors. In addition to that, amisulpride binds with similar affinity to 5-HT7 serotonin receptors, with lower affinity to other serotonine receptors subtypes. Cabergoline has a spectrum of targets wider: it binds with similar affinity to 5-HT2B, 5-HT1D, 5-HT2A. This should be mentioned, discussed and affect the title. 

3) The conclusion that D2 dopamine receptors is not justified by the selectivity of ligands. At least D3 can be also concerned. As the drugs were administered orally, than the effects on all brain structures should be considered.

4) There is no information on which phase of menstrual cycle the women were.

Minor:

1) Page 2: During expectation...is it OK?

Reviewer 2 Report

This is a clinical trial exploring the role of dopamine D2 receptors on pain processing in humans. The paper is well-written and of interest for the readers with important clinical and research implications. Several minor changes are proposed.

In the abstract section, the authors describe that the CO2 laser was used in the experiment. Why was CO2 used? What are the advantages of using it?

The EEG source localization identified specific brain regions. Why regions? Localization of the main regions found, would be of interest in the abstract section.

In the introduction section, the authors are reporting about the analgesic qualities of dopamine. More references are needed. Are there qualities depending on the type of pain?

The authors used Cabergoline (D2R agonist) and amisulpride (D2R antagonist) which both have a high affinity for D2Rs. Do they have the same affinity or similar, or there are significant differences? It should be mentioned.

Did the authors find any gender difference in sensory integration and acute perception?

The conclusions section is really brief. I would expand it.

What can be concluded from the administration of cabergoline, amilsulpride and socium chloride as a whole, and separately?

Round 2

Reviewer 1 Report

All points were resolved